# Activating Specific Handball’s Defensive Motor Behaviors in Young Female Players: A Non-Linear Approach

**DOI:** 10.3390/children10030469

**Published:** 2023-02-27

**Authors:** Sebastián Espoz-Lazo, Claudio Farías-Valenzuela, Claudio Hinojosa-Torres, Frano Giakoni-Ramirez, Pablo Del Val-Martín, Daniel Duclos-Bastías, Pedro Valdivia-Moral

**Affiliations:** 1Facultad de Ciencias para el Cuidado de la Salud, Universidad San Sebastian, Lota 2465, Providencia 7510157, Chile; 2Instituto del Deporte, Universidad de las Américas, Santiago 9170022, Chile; 3Facultad de Ciencias de la Actividad Física y del Deporte, Universidad de Playa Ancha, Valparaíso 2360072, Chile; 4Faculty of Education and Social Sciences, Universidad Andres Bello, Las Condes, Santiago 7550000, Chile; 5Escuela de Educación Física, Pontificia Universidad Católica de Valparaíso, Valparaíso 2374631, Chile; 6Department of Didactics of Musical, Plastic and Corporal Expression, Faculty of Education, University of Granada, 18071 Granada, Spain

**Keywords:** team sport, non-linear pedagogy, methodology, scholar, female

## Abstract

Traditional models to train and teach young players in team sports assume that athletes learn as linear systems. However, an actual methodology called Non-Linear Pedagogy (NLP) accounts for the fact that the players and the team are complex dynamic systems. Experiences in handball under this methodology are scarce; due to this, an observational study has been conducted with a follow-up, idiographic and multidimensional design, in which 14 female school handball players belonging to four different local teams in Santiago, Chile (age = 15.55 + 0.51) agreed to participate in three special handball training sessions with the use of the NLP methodology where three different constraints were used. Descriptive analysis with the Chi-squared test showed a total of 252 observations where most of the variables were dependent on the constraints (*p* ≤ 0.001). Frequency showed that mainly “Defense in Line of progression” and “Proximal contact” were the most activated variables, followed by “Harassment” and “Deterrence” for all constraints. However, only constraint 2 highly activated two collective motor behaviors, while the rest only did it with individual motor behaviors. It is concluded that the constraints used in training seem to be effective in activating a group of defensive handball motor behaviors, specifically those that are basic for female school handball players.

## 1. Introduction

Handball, as a team sport, requires the domain of tactical knowledge, coordinative skills, physical fitness, mental strength, motor skills, social relationships, leadership, communicative skills, power, endurance, speed and decision-making abilities, among several other variables [1,2]. All of these are part of the dimensions that comprise athletes as human beings and which together must interact harmoniously with each player to help them perform as great as possible as a team [3].

However, traditional teaching models for team sports have only focused on the development of technique, followed by tactical skills and the introduction of the rules of the game, using an isolated context complemented with strength and conditioning training [4]. Once the expected technical-tactical skills are acquired, these are taken into a special training session where a transference methodology is used, which progressively introduces the learned skills into the real game by replicating a group of movements pre-defined by the coach [5], repeating linear mechanic-specific movements, converting the participants into replicative passive players limited in their integral development although highly trained in their physical and technical dimensions [4].

In this sense, traditional training models have pushed the nature of teams sports as a non-cyclical game to a standardized one, where the elements of uncertainty have decreased, improving the dependence on strength and conditioning work with a lack of creativity from the players, except for those exceptional highly talented ones [6].

The aforementioned is possible to see in handball, as it has been stated that performance indicators in the World Handball Championship for men and women are exclusively the “Anthropometric Indicators” (only for men), “Age index,” and “Scoring index,” while tactical behaviors, decision-making, and team players interaction are not taken into account [6]. Also, it has been described that handball performance is highly dependent on fast breaks and counter attacks overall other attacking elements. This tactical behavior is described as a cyclic conduct for most female teams, which explains the reduced interest of viewers worldwide compared to men’s handball games [7]. Other studies have shown that for handball, the most relevant elements to training for improving performance are goalkeeper efficacy and distance shooting efficacy in order to determine whether to win or lose a match [8]. However, a recent study from Flores–Rodriguez & Alvite-de-Pablo [9] has demonstrated that the dynamic nature of offensive behaviors is highly relevant to determining goal efficacy. 

In response to these limitations, an alternative teaching model has been developed based on the complex dynamic system theory called Non-Liner Pedagogy (NLP), which involves the interaction between the players and the environment through the implementation of adapted games [10], comprehending that this provides an ambience where players must interrelate between them to solve tactical problems by using all their skills, mainly in an integrated way from the individual perspective as well as the team perspective as a whole and not as a sum of its parts [11].

The NLP allows players to elaborate paths of verbal and non-verbal communication, organize strategical responses, reproduce emergent tactical motor behaviors, analyze the environment to make decisions, and especially adapt their individual resources to improve the team’s performance [12]. The NLP understands that team sports behave as a complex dynamic system, where the team is a dynamic network of players who will interact under the logic of the theory of chaos [13]. This means that every response from the team to a given problem (defensive and/or offensive) will be generated by the interaction of every involved player but also by the value of the interaction itself, given through the perception of each involved player. Thus, every conduct of the team in the field cannot be entirely predicted because multiple variables could affect them [14]. Nonetheless, as a complex dynamic system, there are constraints that push the system to act with certain specific behaviors, although with spontaneous changes of them when something disturbs the system. This change is called the conduct of emergency, which last for a reduced period of time until the disturbance is solved [11].

The NLP to teach team sports was first described by the study of Chow et al. [13] as an argument of the methodology called TGfU (Teaching Games for Understanding), which later evolved into the current NLP as a proper methodology [15]. The arguments presented refer to the use of adapted games where the goal is given, and specific tasks are limited or obligated through what is known as constraints manipulations (CM). This is the first study to introduce the concepts of the complex dynamic system theory in the field of team sports teaching, particularly the use of CM as the key to generating motor behaviors as a solution to the emerging difficulties that constantly appear in the context where the team is involved [13].

With regards to the CMs and their effects on motor behavior, one of the first studies that have described an experimental design is the study of Praxedes et al. [12]. Their research has shown that in football, through a two-vs-one practice, when the playing area is delimited and the players are asked to move and get away from their direct opponent, the players tend to provide a better chance to their teammates to score by moving the defense, this is the first evidence of the higher effectiveness of NLP compared with the traditional model.

Later, specifically in handball, only a few studies were conducted using the NLP [16,17,18,19]. Most of these describe the effects of constraints manipulation on the tactical motor behavior, mainly on the attack, in the context of specific modified game situations, particularly in male players where the use of specific special rules, the limitation of the space, and the use of a specific number of defense lines, impulses desired motor behaviors such as the relation between players to attack by penetrating the defense using crosses, short passes, and blocking.

Limited evidence has been reported according to handball’s defensive motor behaviors developed by NLP. At the time this article was made, only the study of Flores–Rodríguez and Ramírez–Macías [18] has given some specific results of CM on defenders. This study has shown emerging motor behaviors such as individual defense to pivot player, increased number of intended pass interceptions, opponent changes, and counter blocking, among others.

As the evidence has demonstrated, the effectiveness of NLP on handball´s attacking motor behaviosrs is still limited, and due to the lack of more evidence regarding handball’s defensive motor behavior and the absence of this kind of study in female players, researchers have hypothesized that the use of NLP in handball training might activate specific handball’s defensive behaviors. Therefore, is the aim of this research to describe the effects of the use of NLP methodology in the development of defensive motor behaviors in young female handball players.

## 2. Materials and Methods

This is a descriptive, quantitative, observational study planned to obtain relevant information regarding our investigation objective. Particularly, this corresponds to a follow-up, idiographic and multidimensional design, as there was an inter and intra-sessional analysis which provided data about the frequency of the performed actions and about some behaviors of the players independently as part of a system (team) in both individual and collective responses [20]. The sample included 14 female school handball players belonging to 4 different local teams in Santiago, Chile (age = 15.55 + 0.51). All of them were selected by convenience to participate in a special training program for talent identification according to the Chilean National Federation of Handball criteria. Their handball experience comprised systematic training in their clubs for the last 6 years, although the last year was interrupted due to the COVID-19 pandemic, where most of their training was focused on physical and recreational activities through virtual meetings. Two of the players have also experienced playing on the national team in a Pan-American tournament during the year 2020 that included intensive double training sessions during a week every month before the beginning of the tournament. As part of the talent identification program, all the selected players showed handball’s basic motor skills already developed, however, only in terms of individual offensive abilities. At the moment of the intervention, the players had experience with basic handball’s defensive tactical behaviors: ball interception, blockage, and ball interception during dribbling. However, these were technically not efficient as expected. In terms of training load, all players had 2 sessions of handball training in their clubs. However, during this study, clubs only worked on the technical skills they reported. 

Before the study started, both the parents/guardians and the participants were notified and informed about the procedures, making it clear that their participation was totally voluntary. Also, informed consent was signed by the parents/guardians as well as by the players. It should be noted that this research was approved by the Research Ethics Committee of the University of Granada under the registration number: 2000/CEIH/2021, responsibly ascribing to the standards agreed in the Helsinki agreement [21], specifically regarding research and medical procedures involving children.

### 2.1. Procedure

All the selected players participated in 3 special handball training sessions that comprised the same structure described next:(A)Warm-up: Where all players were asked to jog for 5 min, then to practice freely and individually, for 5 more minutes, all the different abilities they knew, and later in couples, to jog freely, passing the ball between them;(B)Individual defensive technical training: Consisted in different exercises related to defensive movements, specifically to move oriented according to their opponent, to maintain distance, or to apply pressure depending on the attacker’s intention;(C)NLP exercises: A specific exercise is applied in every session, however, with different constraints in each 1;(D)Offensive training: Comprised of specific exercises to improve mobility and dynamism during the ball circulation performed by the attackers. The goal was always to keep the speed of the ball, passing it among players while they were moving side to side by performing different combinations of movements with no opponents;(E)Cool-down: In all sessions, players were asked to apply self-massage and stretching.

All participants were asked not to eat food and not to ingest stimulant beverages such as coffee or energy drinks at least an hour before training sessions. Also, players were asked to maintain the same type of diet that they present daily. For hydration, the players did its ad-libitum. At the time players participated in the study training sessions, no training on their clubs was performed, thus not competing either.

### 2.2. NLP Exercise

As seen in previous studies regarding NLP methodology [22,23,24], a specific structure where the players had to interact was given. All the players were distributed according to the coach´s perspective to perform first as attacker or defender, to later change their role to the opposite one. This intended way to distribute players was made in order to balance both groups with similar levels of performance. With this, two teams were organized, the attacker team (AT) and the defender team (DT), both with the same number of players. The AT had 6 handball balls, which had to be left inside the central circle (Figure 1). For this, AT players were able to use all that the rules of handball allow and forbid them to do. On the other hand, the DT had to prevent AT from reaching its goal by using different actions also limited by the rules of handball. The exercise had to be performed 6 times; each repetition had to start every time an attacker achieved the goal or if the attacker lost the ball.

In each session, before the exercise started, the coach explained out loud to the defenders the next: -Defenders must prevent attackers from moving forward to the goal area;-Defenders have individual responsibilities, although they can help their teammates without losing their individual opponents;-Defenders must prevent attackers from penetrating the space between and behind the defenders.

These 3 indications correspond to what Anton (2003) has described as the fundamental pillars of defensive tasks.

For each session, a constraint was given to conditioning the motor behavior of defenders. These constraints were the next:-Session 1: Each attack of the 6 had to be done with the 6 balls at the same time, and no circulation of an attacker could be done through the central circle.-Session 2: Each attack of the 6 had to be done with only one ball at a time, and no circulation of an attacker could be done through the central circle.-Session 3: Each attack of the 6 had to be done with only one ball at a time, and the circulation of an attacker can be done through the central circle as a pivot position for only 3 s.

The coach remembered out loud the constraints for the DT players, and no other information was given before or during the exercise, as well as no feedback after it. It is important to highlight that none of the players had experience with collective defensive tactics before this study.

### 2.3. Data Collection

As this is an observational study, every session was video recorded using the main integrated camera of an iPhone XR 64GB^®^ (USA) added to a tripod which was located on the side of the court 4 m high [25]. Every recorded session was saved in a virtual hard drive for the posterior edition. Every video was edited using Adobe Premiere Pro CC 2020^®^ (USA), in which the NLP exercise of every session was saved independently of the rest of the session. Every NLP video was at the time edited to highlight every time the exercise started or finished and also when the role of the teams changed; this was to facilitate posterior analysis using the Kinovea^®^ software (Open Source). For the observation, an ad hoc observation instrument described by Anguera et al. [21] was adapted using the individual and collective defensive behaviors concepts defined by Anton [26] (pp. 140–155) and completed by Anton [27] (pp. 143–150) (Table 1). This instrument contemplated a system of exhaustive non-exclusive categories that give theoretical support to the observation.

A group of 3 observers, all of them handball coaches, were trained to analyze handball´s individual and collective defensive skills through the observational methodology. A professional handball expert with experience in observational methodology research performed the training for the observers.

As an initial phase, the expert gathered with the observers to consider and clarify any doubts they may have had about the observation process. A pilot video was given to the observers to practice the observational methodology in which the results of the observations were similar in all three coaches.

In the second phase, a video of sessions 1, 2, and 3 were given to each observer in a different order to be analyzed. In this way, each observer got a different sequence of sessions to reduce the influence of the original structured order.

As a third phase, each observer had to qualify the player’s behaviors in each observed situation as NP (Non-Performed): Every time the player should perform the action, but she didn´t; EP (Effectively Performed): Every time the player performed a correct action as the situation need it; NN (Non-Necessary): Every time a defensive behavior was not necessary to be performed. 

Finally, all the results were arranged in an Excel (USA)^®^ spreadsheet using a numeric code for each qualification (NP = 0; EP = 1, NN = 2) to determine the final amount of observation correctly performed.

### 2.4. Statistical Analysis

The statistical software SPSS^®^ v28 (IBM, Chicago, IL, USA) was used for the data analysis. As this is an ideo-graphic, follow-up, and multidimensional observational study, descriptive statistics are presented as frequency. To compare the frequency of the emergent or non-emergent handball´s defensive behaviors regarding the applied constraints, a contingency table was used. To determine if emergent or non-emergent handball motor behaviors are dependent on the applied constraints, the chi-square test was used.

## 3. Results

The descriptive analysis (Table 2 and Table 3) followed by the Chi-squared test (Table 4 and Table 5) has shown a total amount of 252 observations regarding 36 game situations (AT against DT) performed in three sessions in which six attacks per group were performed. For most of the variables, a *p*-value < 0.001 was obtained except for “Proximal Contact,” in which the results showed a *p*-value of <0.005, and for “Defensive Adjustment Through the Back” where the values remain all the same for each observation, so this variable was taken as a constant, so the frequency was not given. In this sense, it is evident that almost all variables are effectively dependent on the constraints. 

When constraint 1 was applied (all attackers with a ball), all the handball’s specific defensive motor behaviors got activated. However, not all of them were at the same frequency. Mainly, two individual behaviors, which are “Defense in line of progression” and “Proximal contact,” were activated the most, with an average of 57 observed actions. For the rest of the handball’s specific defensive motor behaviors, an average of 10 observed actions were obtained (Figure 2).

Regarding constraint 2 (Only one attacker with a ball), “Defense in line of progression” and the “Proximal contact” were highly activated again with the same average of 57 observed actions. Nonetheless, group defensive motor behaviors were also activated in a higher frequency, with 57 and 52 observed actions for “Timely Helps” and Defensive Titl” correspondingly. All the remaining handball’s specific defensive motor behaviors were activated as well but at a lower frequency. However, compared to constraint 1, a higher number of actions were seen for “Harassment”, with 39 more actions in constraint 2 compared with constraint 1. In the same way for “Deterrence”, 24 actions were detected above the four seen in constraint one. A similar phenomenon occurred for “Defensive Adjustment”, “Ball Interception”, and “Change of Opposition”, where an average of 14 observed actions got activated above the seven actions seen during the first constraint (Figure 3).

On the other hand, the third constraint (Only one attacker with a ball plus one pivot) also activated the “Defense in Line of Progression” and “Proximal Contact” in this opportunity with a higher number of observed actions (79 and 74 correspondingly). Differences are possible to observe when comparing constraint three with constraints one and two, as the group defensive motor behaviors reduced their frequency to 16 for “Timely Helps” and 0 for “Defensive Tilt”. Another difference is that “Harassment”, “Deterrence”, and “Defensive Adjustment” were activated more during the third constraint compared with the second and the first. These handball’s defensive motor behaviors presented an average frequency of 23 observed actions above the same behaviors during constraint 2 and an average of 46 above constraint one (Figure 4).

Finally, constraint 2 was the one which has activated more collective actions such as “Timely helps”, “Defensive Tilt”, “Change of Oppositions”, “Deterrence”, and “Defensive Adjustment” compared with constraints 1 and 3. However, constraint 3 has activated in a higher frequency two of the aforementioned behaviors while constraint 1 only activated a few behaviors in a lower amount.

## 4. Discussion

The aim of the present study was to describe the effects of the use of NLP methodology in the development of defensive motor behaviors in young female handball players. For this, three constraints were introduced to a specific exercise which was repeated in three training sessions (one constraint per session) following the examples of the studies of Flores-Rodríguez & Ramírez-Macías [19].

It has been stated that it is essential for handball coaches to be aware of the influence that constraints exert on the players’ behaviors [28]. As mentioned before, three constraints were used in the present study linked to essential and basic individual and collective handball’s offensive actions, such as the one-against-one, passes, unmarking, feints, and players’ circulations, among others, in order to elicit defensive behaviors as the response of these stimuli [27,29].

In handball, to defend against any offensive action, there are several technics and tactical skills which respond to three basic pillars that always have to be accomplished during the whole game: to avoid attackers from getting near their zone of effectiveness, to avoid attackers from shooting at the goal, and to recover the ball at any given opportunity [30,31,32]. The results found in the present study are coincident with the aforementioned, as the specific handball motor behaviors that were activated the most with the three different constraints were “Defense in Line of Progression” and “Proximal Contact,” which are the basic individual conducts that the defenders must have in order to defend the attacker to avoid them to get near to their zone of effectiveness [32]. However, a difference is given in the third constraint, where these motor behaviors were activated more compared with constraints 1 and 2, where the frequency of activation was the same for both (57 observed actions). This was probably given due to the participation of a player as the role of the pivot, who was able to move through the central zone, where the attackers had to leave the ball to accomplish the goal. In this sense, as far the attackers were from the pivot to give her a pass, the higher the possibility of the defenders avoiding the attacker to achieve their goal [33,34,35].

Regarding the motor behaviors of “Harassment” and “Deterrence,” these were slightly activated during constraint one. This may have happened due to the lack of receivers of a pass, as all attackers had a ball in their possession at the beginning of the exercise. In addition, the limitations that attackers in possession of the ball have in terms of mobility throughout the court compared with defenders due to the rules of the game [31] probably had impulse these lasts to stay near the central zone waiting for the attackers to use their cycle of steps to approach, as they couldn’t throw the ball but just leave it on the ground in the central zone. However, in constraints two and three, these motor behaviors increased their frequency of activation as expected due to the addition of attackers without a ball, which allowed at-tackers higher mobility through the court (constraint two) and the addition of a pivot who could enter to the central zone to receive a pass (constraint three). These conducts during all constraints are coincident with the modern defensive systems that are more open and profound, as seen in the latest international handball tournaments when attackers are skilled in distance shooting or closed when the attackers are skillful in the one-against-one but not in the distance shooting [36].

Particularly constraint two was the only one that activated in a high frequency of actions the two collective motor behaviors, “Timely helps” and “Defensive Tilt”. As mentioned before, this constraint allowed more attackers to move unlimitedly through the court, with the exception of the central zone, prioritizing movements in width over those in-depth toward it. In general, the attackers tend to seek free space to penetrate into the central zone, as often is done in the real game in young female teams [37,38]. The logical response to this tactical action is to increase the number of players in the zone where the attacker wants to penetrate in order to persuade the attacker to make a pass or to help in defense when the attacker tries a one-against-one. These defensive responses are commonly proposed in the literature as the basic tactical strategy against the attack. 

In addition, the study of Musa & Menezes [39] about coaches’ opinion on defense skills in young handball players has declared that double marking and staying close together (Timely Helps and Defensive Tilt synonyms) are the basic defensive motor behaviors that young players under 14 must dominate as a prerequisite to training in the subsequent category. In this sense, as the players in our study belong to different U16 teams, it can be expected that in addition to the individual defensive motor behavior used to avoid attackers from reaching the central zone, “Timely helps” and “Defensive Tilt” were performed.

The handball’s motor behavior of “Ball Interception”, “Change of Oppositions”, and “Ball Interception during Dribbling” were all slightly activated in all three constraints. The explanation for this is that in most of the attacks in every session, dribbling was barely used, passes were often given far away from the defense, and crosses of the attacker were not given regularly, except for the constrain two that activate “Change of Opposition” 21 times. However, these crosses were not significant in reaching the goal.

One of the main limitations of our research has been the difficulty of limiting the order effect that sessions may have had on the results. However, as this is an observational study, the results are a description of what has been observed. Also, a second difficulty was finding data regarding technical and tactical skills in young female handball players, as most of the studies refer to strength and conditioning training or anthropometric profiles. However, there was enough literature from well-known international handball coaches that helped the present discussion.

## 5. Conclusions

After the analysis of the results and the discussion of them, the authors agree that the constraints used in the proposed exercise give the impression of being effective in activating a group of specific defensive handball motor behaviors for young female handball players. It is important to highlight that also, by analyzing the three constraints separately, these activate differently, so the exercise used in this research with the correspondent constraint could also be used independently to improve a specific group of defensive motor behaviors.

## Figures and Tables

**Figure 1 children-10-00469-f001:**
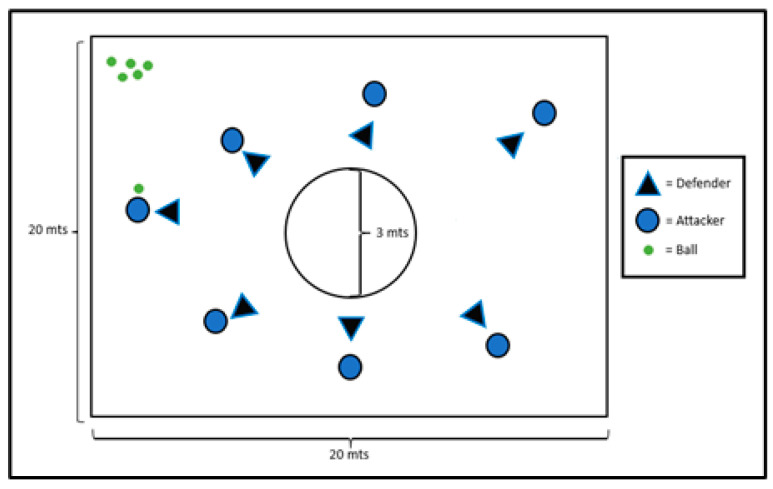
General view of the structure of the NLP exercise.

**Figure 2 children-10-00469-f002:**
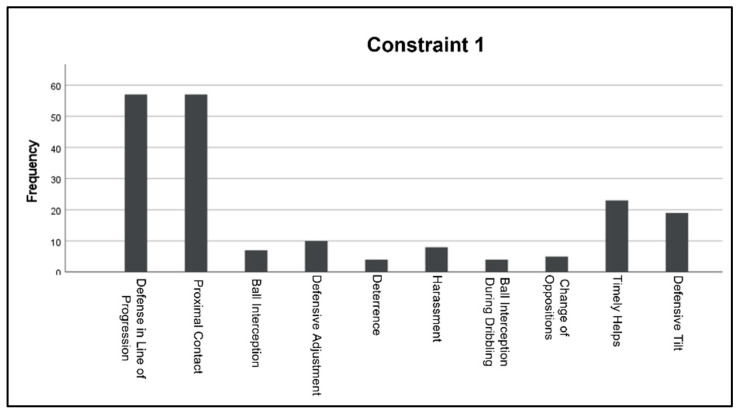
The frequency in which handball’s motor behaviors were activated during the exercise where constraint 1 (All Attackers with one ball) was applied.

**Figure 3 children-10-00469-f003:**
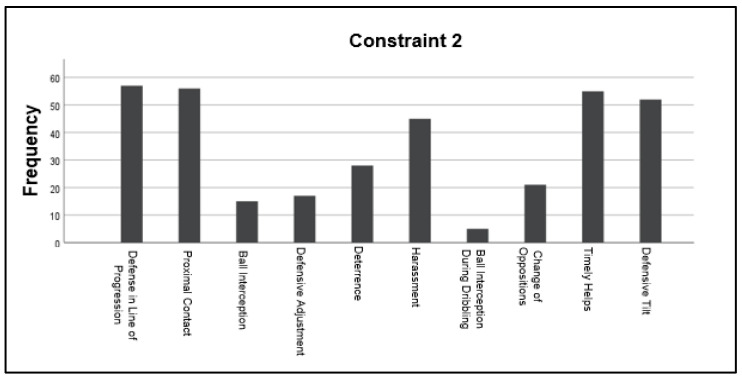
The frequency in which handball’s motor behaviors were activated during the exercise where constraint 2 (One attacker with one ball) was applied.

**Figure 4 children-10-00469-f004:**
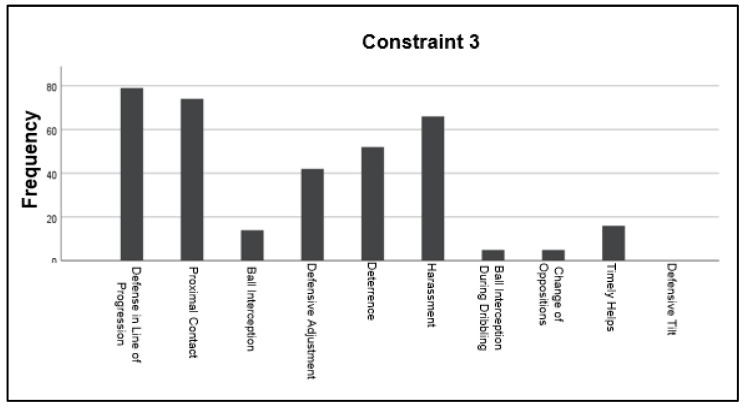
The frequency in which handball’s motor behaviors were activated during the exercise where constraint 3 (One attacker with one ball plus one pivot) was applied.

**Table 1 children-10-00469-t001:** Observational instrument.

Motor Behavior	Description
Defense in Line of Progression	The defender moves in all the directions necessary to stay in the line of progression of her direct opponent towards the arc.
Proximal Contact	The defender contacts her direct opponent with her hands, managing to avoid or hinder, without committing a foul, her progression to the goal.
Ball Interception	The defender orients and moves through space, managing to intercept passes and/or shots on goal.
Defensive Adjustment	The defender observes that the opponent is in an advantageous place to score a goal, so she abandons her mark, which is in a less advantageous place, and immediately assumes the defense of the player who generates more danger of converting a goal, placing herself next to
Deterrence	Marking of the pass line by the defender with the aim of making doubt the possibility of passing and trying to intercept it if it is executed.
Harassment	The defender is located at a very close distance that avoids any type of displacement of the attacker or the possibility of receiving a pass
Ball Interception During Dribbling	The Player observes the cycle of steps and identifies the moment in which the attacker will bounce the ball in order to intercept it during the trajectory of this towards the ground or towards the hand, managing to make the attacker lose control of it.
Change of Opposition	Defenders make changes to their direct opponents, effectively coordinating that each one is marking the direct opponent of their partner, maintaining the lines of progression corresponding to their new marks in case of crossing or blockages.
Timely Helps	The close defender collaborates with her partner, who finds it difficult to defend an attacker who is ready to overcome, managing to defend the attacker who loses the continuity of her action. The collaborating defender does not arrive before or after the moment declared.
Defensive Adjustment Through the Back	The defender in front of a cross in-depth observes that her fellow defender is contacting or in the process of contacting close to her opponent who participates in the crossing, moves to keep her direct opponent in front of herself, passing through the back of her fellow defender, and then recover the defense in line of progression, as long as your partner doesn’t need help.
Defensive Tilt	Displacement of the group to maintain a greater volume of defenders compared to the attackers in order to avoid leaving spaces for the progression towards the goal.

**Table 2 children-10-00469-t002:** Descriptive Analysis of observed Actions regarding Handball’s Motor Behaviors (a).

	Defense in Line of Progression			Proximal Contact			Ball Interception			Defensive Adjustment			Deterrence			Harassment		
	NP	EP	NN	NP	EP	NN	NP	EP	NN	NP	EP	NN	NP	EP	NN	NP	EP	NN
Constraint 1	27	57	0	25	57	2	19	7	58	27	10	47	17	4	63	74	8	2
Constraint 2	22	62	0	28	56	0	37	15	32	22	17	45	56	28	0	39	45	0
Constraint 3	5	79	0	10	74	0	7	14	63	15	42	27	0	52	32	18	66	0

NP = Non-Performed; EP = Effectively Performed; NN = Not Necessary.

**Table 3 children-10-00469-t003:** Descriptive Analysis of observed Actions regarding Handball’s Motor Behaviors (b).

	Ball Interception during Dribbling			Change of Opposition			Timely Helps			Defensive Adjustment through the Back			Defensive Tilt		
	NP	EP	NN	NP	EP	NN	NP	EP	NN	NP	EP	NN	NP	EP	NN
Constraint 1	37	4	43	3	5	76	24	23	37	0	0	84	30	19	35
Constraint 2	4	5	75	34	21	29	19	55	10	0	0	84	12	52	20
Constraint 3	0	5	79	4	5	75	0	16	68	0	0	84	0	0	84

NP = Non-Performed; EP = Effectively Performed; NN = Not Necessary.

**Table 4 children-10-00469-t004:** Frequency Table and Chi-Square Test (a).

	Defense in Line of Progression		Proximal Contact		Ball Interception		Defensive Adjustment		Deterrence		Harassment	
	Value	Sig.	Value	Sig.	Value	Sig.	Value	Sig.	Value	Sig.	Value	Sig.
Pearson’s Chi-square	18.81	<0.001	16.14	<0.003	35.744	<0.001	34.133	<0.001	171.6	<0.001	84.13	<0.001
Likelihood ratio	21.86	<0.001	17.58	<0.001	37.91	<0.001	33.68	<0.001	217.3	<0.001	94.69	<0.001
Linear by linear association	17.04	<0.001	5.026	<0.025	2.348	0.125	0.559	0.455	1.763	0.184	65.06	<0.001
Number of valid cases	252		252		252		252		252		252	

Sig.: Bilateral asymptotic sigma.

**Table 5 children-10-00469-t005:** Frequency Table and Chi-Square Test (b).

	Ball Interception during Dribbling		Change of Opposition		Timely Helps		Defensive Adjustment through the Back	Defensive Tilt	
	Value	Sig.	Value	Sig.	Value	Sig.	It has not been calculated, as SPSS was taken as a constant	Value	Sig.
Pearson’s Chi-square	72.34	<0.001	85.96	<0.001	93.91	<0.001		139.438	<0.001
Likelihood ratio	76.75	<0.001	85.55	<0.001	110.1	<0.001		162.3	<0.001
Linear-by-linear association	56.3	<0.001	0.041	0.839	32.89	<0.001		64.9	<0.001
Number of valid cases	252		252		252			252	

Sig.: Bilateral asymptotic sigma.

## Data Availability

Data is unavailable due to privacy or ethical restrictions regarding children’s data protection.

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
