# Peer review of "Activating Specific Handball’s Defensive Motor Behaviors in Young Female Players: A Non-Linear Approach"

_children, 2023, doi:10.3390/children10030469_

Round 1
Reviewer 1 Report
Dear authors,
please take a look at the comments.

Author Response
Dear reviewer,
Thank you so much for your comments. Next, you will find our poin-by-point response.
Introduction:
After Reading again our introduction we feel that the rationale is well explained. It has an important theoretical background and it is lack of hypothesis as is an observational analysis.
If you insist that is compulsory to add more information regarding the rationale, we will do it.
Method:
The iPhone, Adobe Premiere and excel are from the United States, while Kinova software is an open-source software so has no country of origin. We have added country information for both in the article.
With regards of the addition of graphics, we have already 4, if we include more (at least 3 for each constraints), it will be too many. Thank you anyway for your recommendation.
Results:
Regarding p-value, now is written in italic.
With regards to adding values to the graphs, these are already in the Y axis and also in tables 2 and 3. Do you think is a good idea to add the value also in the bar of the graph? If you do, we will put them there.
Thank you again for helping us to improve our article.
Kind regards.
Reviewer 2 Report
The article is new and explicitly presents the new NPL method, it should be remembered. Congratulations to the authors!Author Response
Dear Reviewer,
Thank you so much for your kind comments.
Regarding the English language check, we have done it, and improved it.
Kind regards.
Reviewer 3 Report
The scientific value of the work is revealed. I don't see any major flaws, but the statement in lines 123-124 "At the moment of the intervention, none of the players had experience with handball's specific defensive tactical behaviors" raises doubts. In the training group, while improving offensive motor skills, defensive skills are usually also being improved. Maybe it is possible to indicate which specific defense skills were not taught.
I would recommend rewriting the conclusions and answering the raised aim of the study, which is not related to separate constraints but to the overall effectiveness of the NLP methodology. The results are now replicated in the conclusions.
Author Response
Dear reviewer,
Thank you so much for your comments. Next, you will find our poin-by-point response.
Regarding your comments about lines 123 and 124. We have changed the statement and clarified that 3 basic defensive behaviours were known by the players previously to the research. However, these were not efficient as spected.
Regarding the conclusions, we have improved by rewriting it according to your suggestions.
Thank you again for helping us to improve our work.